# Human Activity Recording Based on Skin-Strain-Actuated Microfluidic Pumping in Asymmetrically Designed Micro-Channels

**DOI:** 10.3390/s24134207

**Published:** 2024-06-28

**Authors:** Caroline Barbar Askar, Nick Cmager, Rana Altay, I. Emre Araci

**Affiliations:** Department of Bioengineering, Santa Clara University, Santa Clara, CA 95053, USA; cbarbaraskar@scu.edu (C.B.A.); ncmager@scu.edu (N.C.); raltay@scu.edu (R.A.)

**Keywords:** wearables, microfluidics, image-based sensor, pump, human movement continuous monitoring

## Abstract

The capability to record data in passive, image-based wearable sensors can simplify data readouts and eliminate the requirement for the integration of electronic components on the skin. Here, we developed a skin-strain-actuated microfluidic pump (SAMP) that utilizes asymmetric aspect ratio channels for the recording of human activity in the fluidic domain. An analytical model describing the SAMP’s operation mechanism as a wearable microfluidic device was established. Fabrication of the SAMP was achieved using soft lithography from polydimethylsiloxane (PDMS). Benchtop experimental results and theoretical predictions were shown to be in good agreement. The SAMP was mounted on human skin and experiments conducted on volunteer subjects demonstrated the SAMP’s capability to record human activity for hundreds of cycles in the fluidic domain through the observation of a stable liquid meniscus. Proof-of-concept experiments further revealed that the SAMP could quantify a single wrist activity repetition or distinguish between three different shoulder activities.

## 1. Introduction

In recent years, there has been considerable interest in the development of skin-mountable strain sensors (SMSSs) for human movement tracking [1,2,3,4,5,6,7]. These sensors can particularly help with the monitoring of rehabilitation programs for patients suffering from musculoskeletal diseases [7,8,9] or exercise tracking for sports analytics applications [10,11,12,13]. Several strategies were developed to improve the sensitivity (i.e., gauge factor (GF)), stretchability, and linearity, as well as to reduce the hysteresis of SMSSs [1,12,14,15,16,17,18,19]. Capillaric SMSSs developed in our group utilize fluid flow to enhance the sensitivity of strain sensing, as well as to enable integrated sensor networks toward movement recognition [17,18].

Some of the attractive features of SMSSs are their ability to detect skin deformations due to the movement of the joint, as well as the muscle, and to provide continuous data [20,21,22]. Typically, near-field communication (NFC) [23,24,25] or Bluetooth integration has been used for a continuous readout [26,27]. The wireless transfer of electrical signals with such methods requires relatively complex and costly flexible or rigid electronic components to be integrated into the skin regardless of the sensing technology. A potential solution to eliminate the need for electronics for data transfer is an image-based readout [28]. This method is typically used in colorimetric sweat-sensing applications using microfluidic wearables [29]. In a standard application, the sweat interacts with a chromogenic target, which changes color after a chemical reaction [30,31]. The color change can be recorded with a smartphone [32,33,34,35,36,37,38]. Another wearable device technology that relies on an image-based readout is intraocular pressure (IOP) sensing in microfluidic contact lenses. In this case, instead of the color, the liquid meniscus position is measured by a camera to determine the IOP [39,40]. For both applications, the required measurement time intervals are in the order of minutes or larger, eliminating the continuous data transfer need and making the image-based readout a suitable power- and electronics-free method. However, in human-activity-tracking applications, the biomechanical changes occur at the time scale of seconds or faster and are repeated continuously. Therefore, an image-based readout for continuous data transfer cannot be possible without video recordings, which would negate the simplicity advantage of the image-based readout. In some cases, however, the continuous transfer of movement data is not necessary; instead, single-time data that summarizes the activity type or its number of repetitions is sufficient (e.g., fitness or sleep trackers providing daily or weekly activity trends). Summarized data can reduce the data volume and increase the efficiency of processing by reducing the computational load. With a microfluidic device that responds to the aggregated activity, a single image will inform the user of the summary of the activity type, intensity, or repetition. 

Microfluidic skin-mountable strain sensors are capable of detecting small changes in strain using fluidic components, making them highly sensitive for monitoring subtle physiological cues with precision [18]. These sensors offer comfort, as they conform adeptly to the skin’s surface, allowing for prolonged use without causing discomfort or restricting natural movements. Here, we have used the liquid displacement capability in capillaric strain sensors to develop a novel skin-strain-actuated microfluidic pump (SAMP). In this method, the repeated biomechanical changes (e.g., wrist or shoulder movement) cause an accumulated liquid displacement that is measured with a single image of the sensor. To generate a unidirectional flow from cycling applications of external forces (e.g., pressure, strain), the asymmetric flow characteristics (i.e., diodicity) of microchannels are needed [41,42,43]. The asymmetric flow is typically achieved either by components that have moving parts (e.g., one-way valves) [44] or by using nonlinear components (e.g., tesla valves, fluidic rectifiers) [45,46]. The components with moving parts are difficult to fabricate and miniaturize. The nonlinear components, on the other hand, require high flow rates to provide asymmetric flow (i.e., high diodicity) [47]. An SAMP relies on the difference in the deformation characteristics of the microfluidic channels with different aspect ratios (AR = height/width). According to this principle, when strained in the direction orthogonal to the channel elongation, the hydraulic resistance (R) of the high-AR (>1) channels decreases while the R of the low-AR (<1) channels increases, providing asymmetric flow during the cycling application of the strain, independent of the flow rate. This mechanism without any moving parts is easier to fabricate and miniaturize. Reeder et al. showed that such high-AR channels can be used as a finger-pull-triggered valve to initiate pumping; however, the generation of continuous pumping based on periodic skin-strain variations was not demonstrated [48]. Similarly, Li et al. employed high-AR channels to create a single actuation valve that opens under tensile strain for nucleic acid detection [49]. This valve prevented the pre-existing reaction solution from flowing before stretching. Mishra et al. used finger actuation to activate the pump, enabling on-demand sweat analysis by directing the sweat flow [50]. Another valve mechanism operated by tensile strain was presented by Liu et al., where they utilized the mechanical instabilities of the soft materials [51]. By utilizing tunable and reversible micro-patterns, they enabled strain-based active control for programmable microfluidic systems. 

In this study, we present a skin-strain-actuated microfluidic pump (SAMP) that converts cyclic strain into linear liquid flow by utilizing asymmetric flow resistance. An analytical model was developed to calculate the pumping efficiency (PE). The PE is defined as the net volume displacement divided by the total volume displacement per cycle. By employing elastomeric polydimethylsiloxane (PDMS) devices and benchtop experiments, we validated the congruence between the theoretical predictions and the measured PE. Subsequently, experiments were conducted on two volunteers to record the liquid displacement that resulted from repetitive wrist bending. Finally, we used three of the SAMPs on a shoulder to distinguish three different shoulder exercises from each other. By leveraging 3D digital image correlation, the strain on the shoulder was quantified, revealing a correlation between the measured strain and the liquid displacement observed in the volunteer trials. 

## 2. Device Design and Operation Mechanism

The SAMP design is composed of three components: (a) actuator, (b) asymmetric pumping channels (APCs) (i.e., high aspect ratio (AR > 1) versus low AR (<1)), and (c) an optional observation channel, as shown in Figure 1a. The schematic showing different layers of a chip filled with a working ionic liquid (IL) is shown in Figure 1b. When a strain ε in the direction shown is applied, the (i) high-AR and (ii) low-AR pumping channel widths *w* and heights *h* deform, as shown in Figure 1c. When the strain is applied cyclically, the liquid in the observation channels is pumped from the high-AR channel toward the low-AR channel, as depicted Figure 1b in each period of the strain cycle. Here, the actuator is an array of parallel microfluidic channels that expand in volume (i.e., dilatation) under strain orthogonal to its elongation. This principle was previously utilized for physiological (e.g., intraocular pressure (IOP)) and movement-sensing applications in our group. According to the analysis results, when the membrane deformations are negligible (i.e., high spring constant), the actuator can be considered as a cyclic fluid flow source under an applied cyclic strain. Here, we utilized the asymmetry in the strain-induced deformation (ASID) of the APC to convert this cyclic flow into a linear flow. 

The hydraulic resistances *R_LAR_* and *R_HAR_* of the low- and high-aspect-ratio (*AR*) channels, respectively, normalized with respect to the length and viscosity are shown in Equations (1) and (2), respectively [52]:(1)$$R_{LAR}\propto \frac{1}{1-0.63AR}\frac{1}{h^{3}w}$$
(2)$$R_{HAR}\propto \frac{1}{1-0.63{AR}^{-1}}\frac{1}{w^{3}h}$$

When these channels deform, as shown in Figure 1c, the new hydraulic resistances of the deformed channels (*R′_LAR_* and *R′_HAR_*) can be written as follows:(3)$${R^{\prime }}_{LAR}\propto \frac{1}{1-0.63AR^{\prime }}\frac{1}{h^{\prime 3}w^{\prime }}$$
(4)$${R^{\prime }}_{HAR}\propto \frac{1}{1-0.63{AR}^{\prime -1}}\frac{1}{w^{\prime 3}h^{\prime }}$$

Here, $w^{\prime }=w+\varepsilon w$; $h^{\prime }=h-\varepsilon vh$; the aspect ratio $AR=\frac{h}{w}$; and the deformed aspect ratio $AR^{\prime }=\frac{h^{\prime }}{w^{\prime }}=\gamma AR$, where $\gamma =\frac{1-0.5\varepsilon }{1+\varepsilon }$ for Poisson’s ratio v=0.5 [53]. If we assume a small strain and ignore the higher-order terms, the deformed resistances can be written as follows: (5)$${R^{\prime }}_{LAR}\sim R_{LAR}\frac{\left(1-0.63AR\right)}{\left(1-0.63\gamma AR\right)}\frac{1}{\left(1-\frac{\varepsilon }{2}\right)}$$
(6)$${R^{\prime }}_{HAR}\sim R_{HAR}\frac{\left(1-0.63{AR}^{-1}\right)}{\left(1-0.63{\left(\gamma AR\right)}^{-1}\right)}\frac{1}{\left(1+\frac{5\varepsilon }{2}\right)}$$

Here, it can be seen that as the strain increases, the low-AR resistance (Equation (5)) increases because the denominator of both multiplier terms increases, while the high-AR resistance (Equation (6)) decreases, demonstrating that the ASID causes the hydraulic resistances of the low- and high-AR channels to change asymmetrically.

### Equivalent Electrical Circuit Model for Pumping Efficiency Calculation

To theoretically demonstrate how the ASID leads to pumping and calculate the pumping efficiency (PE), we developed a simplified electrical circuit (EEC) model of the APCs in the SAMP. Here, we assumed that a periodic strain function as shown in Figure 2a was applied to the SAMP. When the strain is initially applied at t_o_, the actuator applies a vacuum due to the channel dilatation and creates a flow *Q*^+^*_total_*, as shown in Figure 2b (left). Here, we neglected the time dynamics of the strain application and assumed it was applied in an infinitesimally small time interval. The *Q*^+^*_total_* is the sum of flows from the high-AR and low-AR channels, *Q*^+^*_HAR_* and *Q*^+^*_LAR_*, respectively. As shown in Equations (5) and (6), under strain, the high-AR channel resistance is lower, leading to high flow, whereas the low-AR channel has a higher resistance, leading to lower flow. In the second half of the period, when the strain is released at *t*_1_, the volume of the actuator returns to the original value applying positive pressure, hence creating flow in the opposite direction, which is depicted as *Q*^−^*_total_* in Figure 2b (right). At this point, the resistances of the high- and low-AR channels return to the original values too, distributing the *Q*^−^*_total_* equally into two sides of the pumping channels as *Q*^−^*_HAR_* and *Q*^−^*_LAR_*. Here, we assumed that the initial resistances of the high-AR and low-AR channels were equal; however, this is not a strict requirement to obtain pumping, as demonstrated both theoretically and experimentally in the next sections. The developed EEC model is shown in Figure 2c. As seen, the APCs that are connected to the actuator are represented as two parallel resistors, where we neglected their compliance (i.e., capacitance), as they were much smaller in volume compared with the actuator component. For a comprehensive model of the actuator, including the compliance effects, see Araci et al. [54]. Here, in Figure 2c, we created two models that correspond to the strained (left) and relaxed (right) APCs, assuming steady-state operation in each half of the strain cycle. The dependence of the flow on the resistances during the two halves of the strain cycle can be written as follows:(7)$$Q_{HAR}^{+}=\frac{{R^{\prime }}_{HAR}}{{R^{\prime }}_{LAR}+{R^{\prime }}_{HAR}}Q_{total}^{+}\;;\;\;Q_{LAR}^{+}=\frac{{R^{\prime }}_{LAR}}{{R^{\prime }}_{LAR}+{R^{\prime }}_{HAR}}Q_{total}^{+}$$
(8)$$Q_{HAR}^{-}=\frac{R_{HAR}}{R_{LAR}+R_{HAR}}Q_{total}^{-}\;;\;\;Q_{LAR}^{-}=\frac{R_{LAR}}{R_{LAR}+R_{HAR}}Q_{total}^{-}$$

The difference between the two flow directions gives the net flow in each cycle, as provided by the following equations: (9)$$Q_{HAR}^{NET}=Q_{HAR}^{+}-Q_{HAR}^{-}$$
(10)$$Q_{LAR}^{NET}=Q_{LAR}^{+}-Q_{LAR}^{-}=-Q_{HAR}^{NET}$$

In the fluidic circuit, the flow is a function of time; therefore, instead of measuring the flow, we measured the volume displacement as shown in Figure 2d. The volume displacements can mathematically be expressed as follows: (11)$$V_{HAR}^{NET}=-V_{LAR}^{NET}=\int_{0}^{T}Q_{HAR}^{NET}dt$$

Here, it is notable that the volume displacements in the low-AR and high-AR channels are always equal and opposite in sign providing pumping (i.e., unidirectional flow from the high-AR channel to the low-AR channel) or zero (i.e., no pumping occurs only for identical AR channels). The ratio of *V_net_* to *V_total_* (i.e., total volume displacement provided by the actuator) is defined as the PE and is expressed with the following equation obtained using Equations (7)–(11):(12)$$PE=\frac{{R^{\prime }}_{HAR}}{{R^{\prime }}_{LAR}+{R^{\prime }}_{HAR}}-\frac{R_{HAR}}{R_{LAR}+R_{HAR}}$$

The *PE* is a critical performance parameter that varies with the asymmetry in the pumping channel aspect ratios and strain, independently of the *V_total_*. The *V_total_* can be increased separately from *PE* by making actuators with larger volumes. 

We plotted the theoretical PE for an AR range of 0.1 to 10 at a strain value of 0.1 in Figure 3. As the values of AR were changed, we kept the cross-sectional areas constant. Figure 3a shows the PE for a geometry where the cross-sectional areas were equal for both sides of the pumping channel. In Figure 3b, both the width and height of the right-side channel were increased by a factor of 1.5 to observe the effects of the imbalance in resistances between the left and right sides of the pumping channels. In both cases, we observed that when the two sides had equal ARs, which is marked with the dashed diagonal line, the PE was zero and there was no pumping. In contrast, when the two channels had asymmetrical ARs, which is marked as the solid diagonal line, there was pumping from the high-AR channel to the low-AR channel. Here, the positive sign was assigned to volume displacement from right to left. Therefore, the blue color (i.e., negative values) indicates left-to-right flow. As a result, in all cases, the flow was from the high-AR channel to the low-AR channel. As the AR increased, the PE increased, reaching a maximum of 0.06 when the cross-sectional areas were equal (Figure 3a). When the right-side cross-sectional area was larger (Figure 3b), the maximum PE was lower and appeared at an AR value closer to unity for the right-side channel. This was attributed to the lower resistance of the right side, which resulted in the same *V_net_* without requiring extreme AR values.

## 3. Materials and Methods

### 3.1. Fabrication of Molds 

The molds were designed in Solidworks and then 3D printed using a Formlabs stereolithography printer (Somerville, MA, USA) using Clear Resin. Printing was performed at the highest resolution of 0.025 mm. Printed molds were then washed in a dish filled with Isopropyl Alcohol (Techspray, Kennesaw, GA, USA), which was placed in an ultrasonic bath (VWR International, Radnor, PA, USA) for 45 min to remove residual resin. After air drying, the molds were UV cured using a UVC-1000 device (Hoefer Inc, Holliston, MA, USA) at maximum energy for 30 min. 

### 3.2. Fabrication of Devices 

We modified the standard soft lithography techniques described elsewhere [40] to apply them to be used with 3D-printed molds [55,56,57]. Briefly, for the top layer of PDMS (RTV 615, Momentive, New York, NY, USA), a 10:1 ratio (A:B) of PDMS was degassed in a vacuum for 20 min and then spun onto the 3D-printed mold at 200 rpm, providing a thickness of about 500 µm. The PDMS-coated 3D-printed mold was then subjected to 10 min of degassing in a vacuum to remove bubbles, followed by being placed in an 80 °C oven to cure for 2 h and then punched for inlet/outlet holes. Subsequently, a base layer of 20:1 ratio (A:B) PDMS was spun on a silicon wafer at 140 rpm, resulting in a 700 µm thickness. This base layer was placed in an 80 °C oven for 6 min to partially cure before being thermally bonded to the top layer and placed in an 80 °C oven for 2 h. The choice of thermal bonding is crucial for effective adhesion, given the surface roughness of the 3D-printed mold. Once removed from the oven, the chip was filled with 1-butyl-3-methylimidazolium dicyanamide ([BMIM][N(CN)2]) (viscosity ~28 mPa∙s) containing blue dye for visibility. Finally, plasma was applied to the inlets, followed by a layer of E30CL two-part epoxy (Loctite Inc., Corpus Christi, TX, USA) for sealing. The overall dimensions of the chip were a 30 × 30 mm square that was 1.2 mm in thickness. Devices intended for human testing followed a similar fabrication process but with a lower thickness to reduce the mechanical load of the chip. The human testing devices included a top layer spun at 400 rpm and a bottom layer spun at 700 rpm to achieve an overall thickness of 500 µm. 

### 3.3. Benchtop Characterization

We characterized the fluid movement on the benchtop using a mechanical characterization tool Mach-1 (Biomomentum Inc., Laval, QC, Canada). The experimental process starts with setting the liquid interface at a roughly identical position for HAR and LAR channels and allowing it to settle for 5 min. Then, we applied the cyclic strain and allowed the liquid to settle before making a measurement. The measurements were all performed using ImageJ (version 1.54) [58] by manually measuring the liquid interface positions.

### 3.4. DIC Experimentation Methods

To quantify the skin strain, 3D digital image correlation (DIC) was performed using the open-source MATLAB-based software NCORR (version 1.2.2) [59]. The DuoDIC algorithm, developed by D. Solav, allowed us to accurately measure the skin by using two cameras and consider the z displacement error. The camera system was composed of Raspberry Pi modules and a Raspberry Pi lens. The speckle pattern was applied to the skin using a temporary tattoo (Laser Tattoo Paper, FOREVER, Saint Charles, IL, USA). To control the glare of the speckle pattern, we used a softbox lighting system. We also found that applying baby powder on the tattoo reduced the glare while maintaining an appropriate contrast. The speckle pattern was generated using a Speckle Generator [Correlated Solutions, Irmo, SC, USA]. During imaging, we performed three distinct shoulder movements: abduction, shrug, and row. For each movement, a minimum of 5 images were taken throughout the range of motion. When processing the images, the DIC parameter subset radius was set to 30 and the subset spacing was set to 15. The SAMP locations and reservoir orientations were initially marked on the shoulder with a permanent marker to calculate the strain on the sensors. 

## 4. Results

We completed experiments in two categories: (1) benchtop experiments as a proof of the operation principle of asymmetric pumping channels (APCs) and their performance characterization, and (2) human volunteer experiments to demonstrate the capability to record human activity using the SAMP.

### 4.1. Benchtop Experiments

First, we made a control experiment to show that the asymmetry of the pumping channel AR is a requirement for pumping. We designed three different pumping channel types: (1) symmetrical (i.e., identical channel cross-section), (2) different cross-sections but symmetrical AR, and (3) asymmetrical AR (i.e., one high (>1), one low AR (<1)). We applied cyclic strain (ε = 0.05) 10 times to each device on the Mach-1 as shown in Figure 4a and took a photo after waiting for 2 min and repeated this three times in total for thirty actuations. Due to the low viscosity of the working fluid (~28 mPa∙s), the time constant was around two seconds and stabilization was achieved in about 15 s; therefore, two minutes of waiting time was sufficient to reach a steady state. In each device, we took two volume displacement measurements from two opposite pumping channels, and we assigned opposite signs to the volume displacement of each side of the channel (i.e., positive sign for LAR and negative sign for HAR). We used the net volume displacement to total volume displacement ratio as a performance parameter, as described earlier. Then, we plotted these values for each side of the pumping channel for all three of the device designs. Figure 4b shows the results of this experiment. Here, it is clear that the pumping occurred only for the design with an asymmetric AR, and its direction was from the HAR to the LAR, as predicted by theory. 

To investigate the effect of the strain on the pumping performance, we performed an experiment where we varied the strain on the SAMP. In this case, we actuated the devices at a predetermined strain in a second and released the strain in a second, and then waited for flow stabilization for about one minute. We repeated this four times before taking a photo of the device. We completed five such measurements for each strain value. This measurement was repeated three times and the average of the three experiments was found. Figure 4c shows the results of this experiment. We found the best linear fit for each strain dataset. The slopes of these fit lines gave us the pumping efficiency (PE). It is observed that as the strain increased, the PE increased. This shows that the asymmetric deformation of the APC under uniaxial strain was responsible for the pumping. We repeated this experiment in three pristine chips and performed four different experiments (i.e., Chip 1 was repeated in two different ways). In all cases, we observed an increased PE with increased strain, as suggested by the theory. Figure 4d shows the measured PEs from these experiments (i.e., Chip 1, Chip 1-Random, Chip 2, and Chip 3–10 Cycle) in comparison with the theoretical calculations of the PE (solid lines). Experimentally, we observed a decrease in the linearity and higher noise when the strain values were applied in a random sequence (e.g., Chip 1-Random was tested three times at strain sequences of (1) 6, 9, and 11%; (2) 9, 6, and 11%; and (3) 6, 9, and 11%, and exhibited an R^2^ of 0.6 as opposed to 0.9 when tested sequentially at 6%, 9%, and 11% strains). In addition, when the strain values were applied in 10 continuous cycles, there was a reduction in the linearity, albeit there was no increase in the noise. For the theoretical calculations, we used the experimentally measured channel dimensions shown in the inset of Figure 4b (middle). We observed that when the effect of the observation channel was not included, the experimental PE was at least a factor of four less than the theoretical calculations. However, when the effect of the observation channel was included, the experiments and theory were in much better agreement. This was because the observation channels on each side were identical; therefore, instead of contributing to the pumping, they were detrimental to the PE. The remaining discrepancy between the theory and experiment could have been due to the time dynamics of the application of the strain. For each strain, we applied the strain in one second, as described before. However, in the theoretical calculations, we assumed that the load was applied in an infinitesimally small time duration, therefore neglecting the time dynamics of the load function. Since the strain was applied slowly in the experiment, the liquid flow started at lower strains and the measured PE should be compared with the convolution of the load function and the theoretical strain-dependent PE. 

Then, we repeated the above experiment without waiting for stabilization during the application of the strain, as this is a realistic scenario in human activity tracking. In this case, we applied the strain consecutively 10 times and then waited for stabilization before the measurement. Here, the strain-dependent PE was still observed; however, there was a reduction in the PE and its linearity. The results of this experiment are summarized in Figure 4d (Chip 3, red circles) as well.

Figure 4e illustrates how the dynamics of the cyclic strain influenced the pumping efficiency of the SAMP for the identical strain values of 0.05. All of the data here were obtained on the same chip. When there was a waiting period for stabilization between individual actuations (one cycle), the pumping efficiency was about 0.0051, whereas when 10 cycles were applied back-to-back without any waiting periods, the pumping efficiency reduced. As the number of back-to-back cycles increased to 20, we saw a further decrease in the pumping efficiency to about 0.0014, nearly a factor of three less than when the strain was applied after stabilization. This result can be intuitively understood as the stabilization of the liquid flow allowing the pumping efficiency to reach its full performance. However, when stabilization was not allowed between each actuation, the liquid did not find enough time to complete its cycle, and hence, the reductions in *V_net_* and PE were observed.

When the device underwent hundreds of cycles of strain (ε = 0.05), a reduction in the linearity was observed, as shown in Figure 4f (blue circles). This was due to the additional resistance of the observation channels. As the pumping continued, the amount of liquid in the observation channel of the LAR increased, whereas it decreased for the observation channel in the HAR. This created a further imbalance between the two sides, reducing the PE. To improve the linearity, a new SAMP was designed. Here, the observation channel was eliminated, and the APCs were continuously extended throughout the device. Figure 4f (orange squares) shows the results of an experiment using this new design over the course of 200 actuations. We saw that the continuous design produced significantly more linear results in comparison with the non-continuous design. We also saw a better PE performance from the continuous design (no observation channel), in agreement with the theoretical predictions (see Figure 4d). The PE improvement was not as high as the theory predicted because the HAR side of the pumping channel was 3D printed at a lower AR than the design value, at around an AR of one.

### 4.2. Human Volunteer Experiments

After validating the operation and characterizing the performance of the SAMP on the benchtop experiments, we used the design characterized in Figure 4c for human activity quantification and recognition. We determined two locations for our experiments: (1) wrist and (2) shoulder. We chose the wrist, hypothesizing that the smaller degree of freedom of the wrist would allow for simpler quantification of the wrist bending. We chose the shoulder to test the SAMP’s ability to recognize different exercise types. The shoulder joint is complex, with much higher degrees of freedom compared with the wrist. We hypothesized that exercise-dependent skin deformations on the shoulder can be used as a fingerprint of the exercise. Therefore, three SAMPs were simultaneously used during the shoulder experiments.

#### 4.2.1. Wrist Experiments

The device was adhered to the wrist using glue (SkinTite, Smooth-on Inc., Macungie, PA, USA), and then left to dry for 15 min. The arm lay on its side on a flat surface with a block to ensure the wrist extended the same angle in each trial. The exact location of the device on the wrist was marked to ensure all devices were placed consistently. Three devices were tested for 10 sets of 10 repetitions of the wrist extensions, with 2-min waiting periods between each set to allow the liquid to settle before the readout. The experimental setup and the results are shown in Figure 5a and Figure 5b, respectively. Here, the average of the results of three experiments are shown for two test subjects. The standard deviations were obtained from these three experiments. The best linear fit line was plotted for each subject. For both subjects, a strong correlation between the number of moves and the liquid displaced was observed. Test subject 2 observed a greater volume displacement per trial of 0.0205 µL/bending in comparison with test subject 1, who exhibited 0.011 µL/bending. The data from test subject 2 also had larger error between the measurements. These can be explained by the differences in biomechanical properties of the wrist (e.g., size, flexibility, skin elasticity, etc.) between the two subjects. 

#### 4.2.2. Shoulder Experiments

Due to the complexity of the shoulder, we started the shoulder experiments with strain measurements using digital image correlation (DIC). We obtained the first and second Lagrangian skin strains (Epc1 and Epc2, respectively), as shown in Figure 6a–c for three different shoulder exercises. An interactive MATLAB script was used to outline the SAMP locations and simultaneously plot the Epc1 and Epc2 unit vectors at the SAMP locations, as shown in Figure 6d. Here, the arrow directions represent the strain orientations, and the color gradients represent the strain magnitudes. A straight line orthogonal to the SAMP reservoirs (i.e., sensor orientation) was drawn. The unit vectors were then projected on the sensor orientation, as demonstrated in Figure 6e. The projections were then multiplied by their respective magnitudes (M1, M2). The sum of the two projected vectors represented the resultant strain magnitude at a given point. The average resultant strain magnitude <ε_R_> in the sensor area was calculated from the individual ε_R_ values. 

After the DIC analysis, three SAMP devices were placed on the shoulder at the pre-determined locations. Specifically, Chip A was placed on the proximal end of the biceps brachii muscle. Chip B was placed at the clavicular head of the pectoralis major. Chip C was placed on the sternum head of the pectoralis major lateral from the midline, as shown in Figure 7b. These locations and movements were chosen based on the large strain magnitudes and movement dependent strain variations in these areas [60]. The actuators of the chips were placed closest to the center of the shoulder where the strain was the greatest, forming a delta shape. 

The SAMPs were tested for three exercises: shrug, abduction, and rowing. Each exercise was performed for 10 sets, with each set consisting of 10 repetitions, with 2-min rest intervals between each set given to allow the liquid to settle, similar to the wrist experiment. The photos of the SAMPs were taken at the end of each set. We completed three tests with this procedure using pristine SAMPs each time. Each test involved performing all three exercises but in a different order. Figure 7a shows the average pumping rate per actuation (i.e., actuation is a single repetition of a move) with respect to the average resultant strain magnitude <ε_R_> on the sensor. The average pumping rate was obtained from the slope of the above-described measurements of 10 data. The error bars are the standard deviations of the pumping rates from the three tests. Despite being obtained from different locations and exercises, there was a good correlation (correlation coefficient = 0.86) between the average strain the chips experienced and the liquid volume pumped, indicating that SAMP-based measurement is a viable approach for recording different human movements in the fluidic domain. We used the box plot shown in Figure 7b to summarize our results and observe the movement-dependent pumping response. The median is marked as the horizontal line inside the colored box. The box represents the interquartile range (IQR), with the bottom edge marking the 25th percentile and the top edge marking the 75th percentile. The whiskers stretching vertically outside the box represent the variability of the minimum and maximum in comparison with the IQR. We saw that the abduction (blue boxes) resulted in large positive pumping only in chip location A, whereas rowing (green boxes) resulted in large positive pumping only in chip location C. Distinctly, the shrug (orange boxes) resulted in mostly negative pumping in all locations. These results indicate that our approach is suitable for distinguishing different moves from each other. For conclusive results, there is a need for experiments with a larger number of subjects, exercises, and repetitions. 

## 5. Discussion

In recent years, we have seen a myriad of skin-mountable strain sensors that are designed to provide real-time, accurate strain data with enhanced performances, utilizing advancements in material sciences, nanotechnology, and mechanical engineering [61,62,63,64,65,66,67,68,69,70]. Gauge factors higher than 100 [17,61,62,63,64,65,66], as well as a strain detection limit of 0.05% or better [40,67,68,69,70], have been reported. Compared with these reports, the SAMP strain detection limit is low and was found to be limited at around 3%. However, to transfer or store the data, these sensors require electronic components that are interfaced with the skin. Smartphone-based data transfer represents an attractive solution widely applied in wearable devices for chromogenic assays and intraocular pressure (IOP) tracking. The SAMP applied the smartphone-based data transfer method to human activity tracking for the first time. The SAMP can store the activity data in the form of a fluidic interface as another advantage. Therefore, as opposed to the motion capture that requires continuous recording, SAMP provides information about an activity from a single image.

We obtained good agreement between the equivalent electrical circuit model (EECM) and the benchtop experiments for the pumping efficiencies. As an example, at ε = 0.01, the theoretical PE was 0.05 for an antisymmetric PC combination with AR = 3 and 1/3. Our printed molds were not antisymmetric, and therefore, the theoretical PE was calculated to be 0.04. When the observation channel resistance contributions were included in Equations (1)–(4), the PE decreased to 0.02. Our experimental PE measurements were as high as 0.015, suggesting our model explained the pumping mechanism. This model only considers the strain-dependent hydraulic resistance changes as the source of the pumping mechanism. Even though this hypothesis was supported by our results, it can be argued that other fluidic properties, such as the capillary pressure, affected the net flow, which contributed to the PE variability. In addition, in our simplified EEC model, we assumed the load was uniform and orthogonal to the pumping channels; however, this is not always the case in real-life scenarios. Even for the Mach-1 experiments, the placement of the device could change the load profile. We saw the effects of the non-uniform stresses, especially in the human experiments, as the noise in the pumping rates were much larger. In future studies, the theoretical model should be improved to include the surface energy and the non-uniform load effects for more accurate designs. 

The noise in the human shoulder experiments could also be explained by the variations in the way the moves were being performed. In the shoulder experiments, the subjects performed the experiments in an uncontrolled fashion, and this would lead to variations from actuation to actuation in each set. For better characterization of the SAMP devices during complex activities, on joints with multiple degrees of freedom, in the future, we suggest that they are tested when the subject is performing the moves in a controlled fashion using continuous passive motion (CPM) instruments commonly used for rehabilitation [71].

An important factor that needs to be considered during the use of the SAMP is the detrimental effects of sweat. Excessive sweat when an SAMP is used for long durations of time or for strenuous activities may cause defects (e.g., working liquid contamination or blockages), as the outlets of the observation channels are open for maintaining atmospheric pressure conditions. To remedy this, external surfaces of the outlet areas can be treated to be hydrophobic, reducing the risk of sweat contamination. During the operation of our devices (up to 2 h), we did not observe the detrimental effects of sweat because our activities (i.e., regular lab activities or activities of physical rehabilitation) were not strenuous enough to cause sweating. This scenario (i.e., used for a limited time duration (1–2 h) for tracking the quality and quantity of a physical rehabilitation routine) well represents how we envision the SAMP will be used. Typically, physical therapy involves 4–5 different exercises each being performed as three sets of 10 repetitions. This would mean recording of less than 200 actuations in total, which is within the limits of our dynamic range, as demonstrated in Figure 5 and Figure 7. Additionally, the dynamic range of the device can be extended by optimizing the channel geometry (e.g., longer observation channels will improve the dynamic range). 

We fabricated our molds using 3D printing by SLA to expand the AR values we could obtain. Even though this method provided us with fast prototyping and a larger AR (~2.5), the printing fidelity of the channel cross-sections was low, their shapes were not perfectly rectangle as designed, and the channels had the characteristic roughness observed in 3D printed parts. Such fabrication errors can contribute to noise and deviations from the design goals. For this reason, we plan to use photolithography (PL)-based molds in the future. The AR limitations of PL are not of concern, as the pumping efficiency is still significant for an AR of two when the pumping channels are antisymmetric, based on the presented theoretical and experimental findings. As another advantage of the PL for the SAMP fabrication, PL will enable the fabrication of much thinner SAMPs (<100 µm) that should significantly reduce the mechanical load of the SAMP, and hence, improve the correlation between the skin strain and pumping performance.

## 6. Conclusions

We designed and fabricated a skin-mountable strain-actuated microfluidic device for the image-based tracking of human activity without the requirement for motion capture or flexible electronics. The device operation mechanism is explained through the coupling between the mechanical deformation of elastomeric channels and their hydraulic resistances. We identified the asymmetry in the aspect ratio of channels as a way to control the deformation-induced hydraulic resistance changes. The presented device is a strain-actuated microfluidic pump (SAMP) that converts cyclic strain into linear liquid flow by utilizing the asymmetric flow resistance generated in two halves of the strain cycle without requiring any moving parts or high flow rates. We developed a simplified equivalent electrical circuit model of the asymmetric pumping channels of the device that includes a simple theoretical deformation model of microchannels. The model accurately predicted the device’s performance. Overall, the SAMP acts as a memory device that records strain data. Since human activity generates skin strain, a skin-mounted SAMP records human activity. We demonstrated this on two body positions: the wrist and the shoulder. We showed that the resulting liquid position at the end of an activity period has the potential to inform the user of the type and intensity of an exercise. For the analysis of multiple activities of complicated joints, such as shoulders, there is a need for multiple SAMPs. The unique pumping rate from each of these SAMPs can be measured for each exercise and their combined data would allow for the differentiation of several movements as a future study.

## Figures and Tables

**Figure 1 sensors-24-04207-f001:**
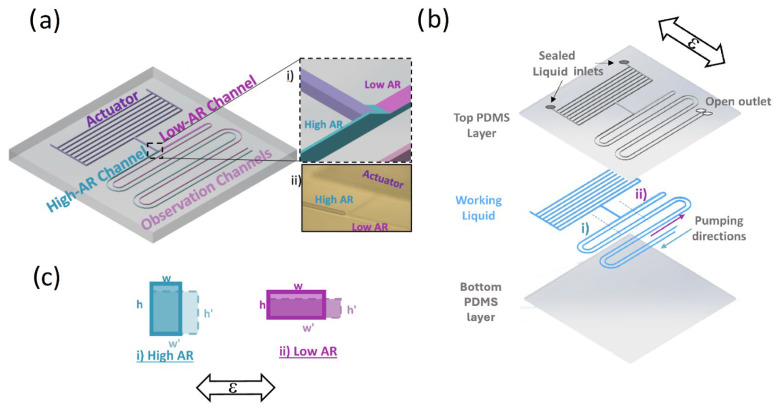
(**a**) A 3-dimensional schematic of the SAMP design, where individual components are labeled. The inset (i) shows the dashed square zoomed in, where the actuator component is connected to the asymmetric AR channels. The inset (ii) is a photo of the same region of the 3D-printed mold. (**b**) A schematic of the flexible chip where individual chip layers are shown separately. The pumping direction is from the high-AR channel toward the low-AR channel as shown. The cross-section schematic of the channels along the dashed line (i) for high AR and (ii) for low AR are shown in (**c**) for before the strain (solid line) and under strain (dashed line). The applied strain ε orientation is shown with bidirectional arrows in (**b**) and (**c**).

**Figure 2 sensors-24-04207-f002:**
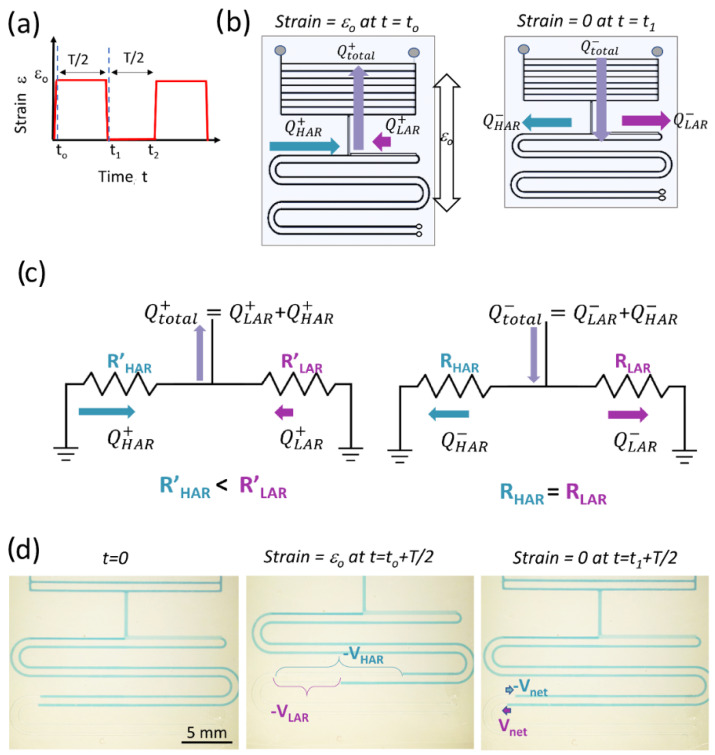
(**a**) The periodic strain function applied to the SAMP. (**b**) A schematic showing the SAMP with arrows representing the fluid flow directions and amplitudes when strain is applied (left) and when strain is released (right). (**c**) The simplified equivalent electrical circuit model of APCs with arrows representing the flow directions and amplitudes when strain is applied (left) and when released (right). (**d**) The photos of the chip before strain is applied at *t* = 0 (left), when strain is applied and held until a steady state (middle), and when strain is released and held until a steady state (right). The total volumetric displacement in each pumping channel after strain is applied and the net displacement after strain is released are indicated.

**Figure 3 sensors-24-04207-f003:**
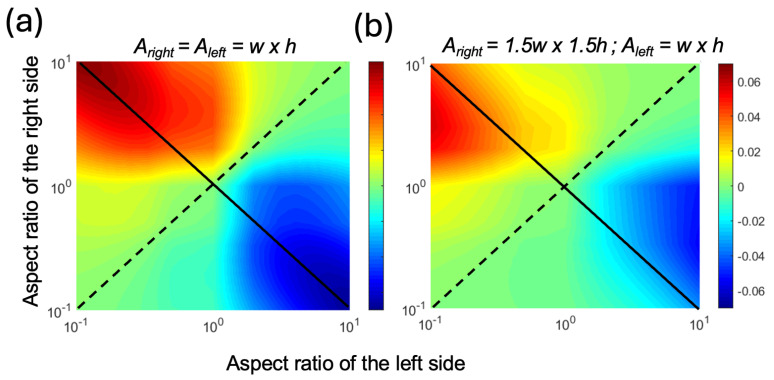
The heat maps of the pumping efficiency (PE) with respect to the varying AR for the left and right pumping channels at a strain ε of 0.1, assuming volume displacement from right to left (**a**) when both sides had identical cross-sectional areas and (**b**) when the right side had a width and height 1.5 times larger than the left side. The solid diagonal line shows the antisymmetric AR combinations, and the dotted diagonal line shows the symmetric AR combinations.

**Figure 4 sensors-24-04207-f004:**
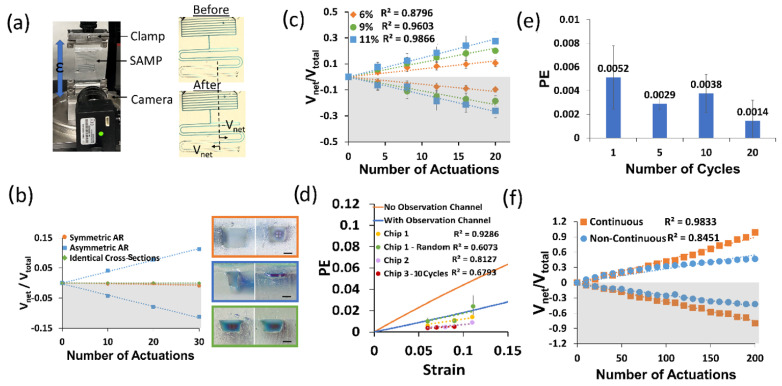
(**a**) A photo of the SAMP test setup and images of the SAMP before (top) and after (bottom) actuation at the strain ε = 0.06 five times. The volume displacement *V_net_* for the LAR was positive and *V_net_* for the HAR was negative. (**b**) The *V_net_*/*V_total_* with respect to the number of SAMP actuations for three different device designs. The pumping channel cross-section photos of each design are shown as an inset (top: symmetrical AR, middle: asymmetrical AR, bottom: identical cross-sections). The scale bars are 200 µm. (**c**) The *V_net_*/*V_total_* with respect to the number of SAMP actuations for varying strain values for the asymmetric AR design shown in (**b**). Three measurements were made on the same chip (i.e., Chip 1) and their averages were plotted. The error bars show the standard deviations of these experiments. The best linear fits and their R^2^ values are shown. (**d**) The strain versus PE calculated using Equation (12) in comparison with the experimental PE (i.e., slope of the lines in (**c**)) for three different chips and four different experiments (i.e., Chip 1 was tested two different ways: increasing strain and randomly applied strain). (**e**) PE measured at 5% strain (ε = 0.05) for varying numbers of continuous strain cycles. (**f**) The *V_net_*/*V_total_* with respect to the number of SAMP actuations for a continuous pumping channel (i.e., no observation channel) in comparison with a non-continuous pumping channel (i.e., with an observation channel). In (**b**,**c**,**f**), the bottom grey shaded (negative values) sides show the HAR channel pumping toward the LAR and the top white areas show the LAR channel pumping toward the outlet.

**Figure 5 sensors-24-04207-f005:**
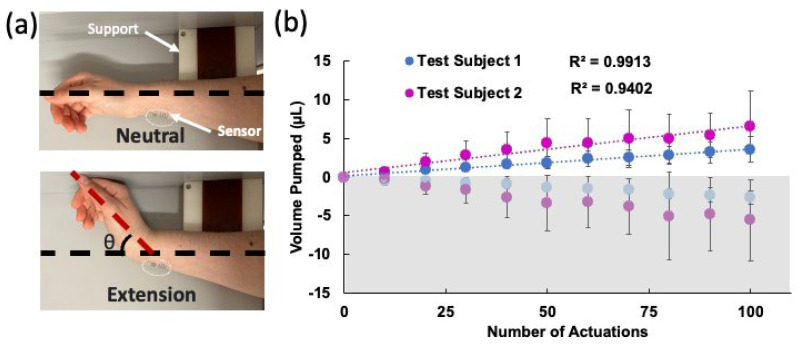
(**a**) Photos of the wrist during the experiment in neutral and extended positions. The support block allowed the wrist to be extended at approximately the same angle each time. (**b**) Volume displacement over the course of 100 actuations for two test subjects. Three different devices were tested on each subject and the averages were plotted. The error bars show the standard deviations of these experiments. The best linear fits and their R^2^ values are also shown. The bottom grey shaded (negative values) side shows the HAR channel pumping toward the LAR and top white area shows the LAR channel pumping toward the outlet.

**Figure 6 sensors-24-04207-f006:**
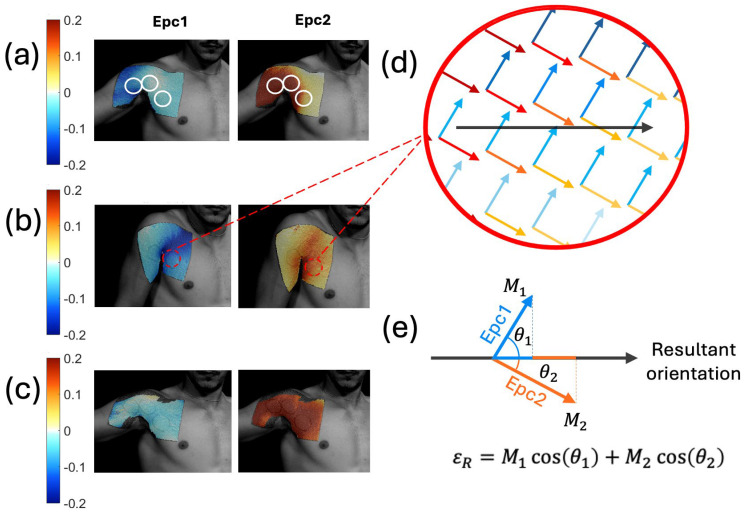
First and second principal Lagrangian skin strains during shoulder movements of an (**a**) abduction, (**b**) shrug, and (**c**) row. The white circle outlined on (**a**) shows the SAMP locations. (**d**) Epc1 and Epc2 unit vectors plotted on the same SAMP location. Here, the blue arrows represent the Epc1 strain vectors; red arrows represent the Epc2 strain vectors. The color gradient shows the strain magnitude values. The black arrow shows the sensor orientation. (**e**) The depiction of the resultant strain magnitude *ε_R_* calculation by adding the Epc1 and Epc2 vector projections on the sensor orientation direction.

**Figure 7 sensors-24-04207-f007:**
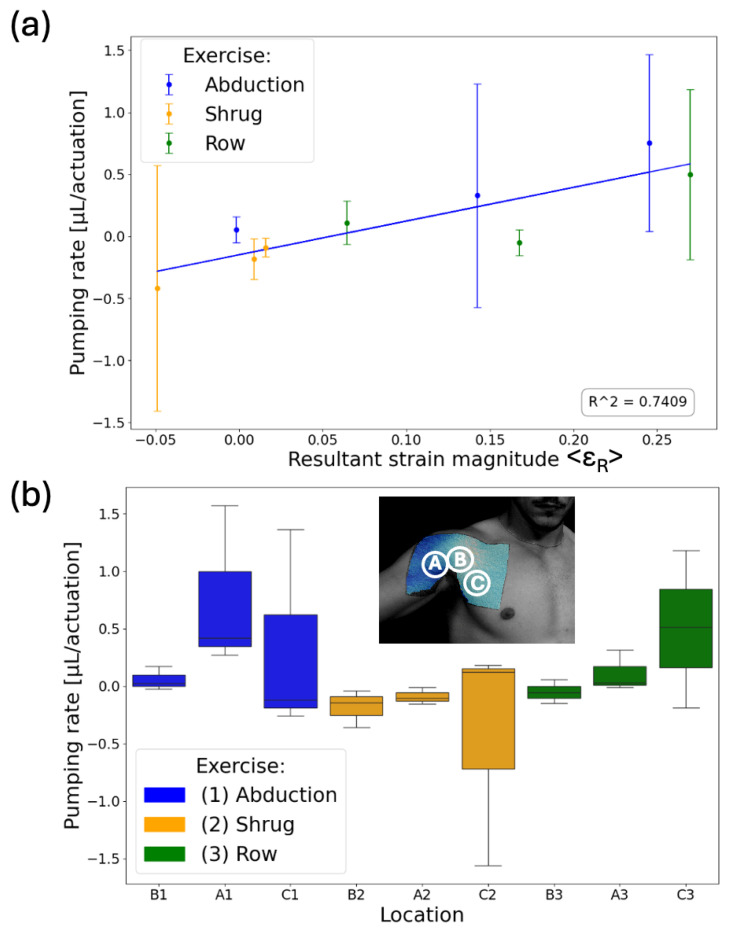
(**a**) Average pumping rate per actuation with respect to average resultant strain magnitude. The SAMPs’ liquid displacements were measured after each set (i.e., 10 repetitions) of exercises and divided by 10 to obtain the pumping rate. The blue line is the best linear fit with an R^2^ value of 0.74. (**b**) A box plot showing the pumping rate dependency on the specific exercise for each SAMP. The letters (A, B, C) represent the location and the numbers (1, 2, 3) represent the type of the exercise.

## Data Availability

The data that support the findings of this study are available from the corresponding author upon reasonable request.

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
