# Peer review of "Human Activity Recording Based on Skin-Strain-Actuated Microfluidic Pumping in Asymmetrically Designed Micro-Channels"

_sensors, 2024, doi:10.3390/s24134207_

Round 1

Reviewer 1 Report

Comments and Suggestions for Authors

In this article, Askar et. al. have presented a skin-strain-actuated microfluidic pump (SAMP) that converts cyclic strain into linear liquid flow by utilizing asymmetric flow resistance, which could be used to detect and measure repeated biomechanical changes. The central idea of this article is to produce asymmetric flow. Typically, asymmetric flow is achieved by components that have moving parts, or by using nonlinear components, which, however, is either complicated to be fabricated and miniaturized or requires high flow rates to produce asymmetric flows. In the article, Askar et. al. have designed a novel pump in purpose of overcoming the intrinsic drawbacks mentioned above.

Having elaborated the working principle of the pump and validated the congruence between theoretical predictions and experiments, Askar et. al. have conducted experiments on volunteers, recording their wrist bending and distinguishing their different shoulder exercises.    

Following are some comments of mine.

Comment 1 

In line 130 to line 144, page 4, it is recommended that some necessary annotations be added following these formulas in order to clarify what RLAR, RHAR, RLAR and RHAR stand for. In the same way, physical quantities such as AR, h, w, and so on are also suggested to be clarified briefly for better readability.

Comment 2 

In Figure 2 d), page 6, the authors have developed an electrical circuit modal of the SAMP, which is a circuitry of pure resistance. An important characteristic of the circuitry of pure resistance is that the phase of response does not delay. Nevertheless, In line 396, page 11, it is known that after each test, 2-minute waiting period before readout is required. Therefore, it is recommended that the authors set up a more complicated circuit model which includes capacitors or inductors to simulate the time-related relaxation process. Moreover, it is suggested that some parameters that may affect the result should be taken into consideration, such as the roughness of the micro-channel or the viscosity of the fluid that flows in the channel.

Comment 3   

 It is recommended that the affect of sweat be taken into consideration because sweat may contaminate the working fluid and affects the accuracy and feasibility of the pump.

Reviewer 2 Report

Comments and Suggestions for Authors

This work introduces a skin-strain actuated microfluidic pump (SAMP) utilizing asymmetric aspect-ratio channels for the recording of human activity in the fluidic domain and established an analytical model to describe the SAMP's operation mechanism. Although this method is interesting and the authors show some useful results, there are some points on which the authors need to provide a more through discussion.

1. From the design of the device and the results of device performance, it seems the advantage of this SAMP for recording human activity is very weak. Although the authors mention SAMP needs no motion capture or flexible electronics, it indeed needs to use a camera to check the position of liquid front. The performance of SAMP is weaker than sensors based on soft electronics, and the authors also mention 'For complicated joints such as shoulders, there is a need for multiple SAMPs'. Therefore, I suggest to the authors to add a detailed comparsion between SAMP and other sensors with similar targets in the discussion, in terms of performance and complexity.

2. SAMP is tested for 'hundreds of cycles'. I suggest the authors to comment on the test cycle numbers to clarify if 'hundreds of cycles' is enough for such sensors based on fluidics instead of electronics. 

3. In Figure 7(a), is R2 value of 0.74 too low? And the error bar seems to be too big, is it a problem for the reliability of this device?

Comments on the Quality of English Language

There are some typos, for example, 'continuos' in Line 82. The authors should check through the manuscript. 
